# Mitochondrial Dysfunction in Cancer Cachexia: Impact on Muscle Health and Regeneration

**DOI:** 10.3390/cells10113150

**Published:** 2021-11-12

**Authors:** Marc Beltrà, Fabrizio Pin, Riccardo Ballarò, Paola Costelli, Fabio Penna

**Affiliations:** 1Department of Clinical and Biological Sciences, University of Torino, 10125 Torino, Italy; marc.beltrabach@unito.it (M.B.); RBallaro@mdanderson.org (R.B.); paola.costelli@unito.it (P.C.); 2Interuniversity Institute of Myology, 61029 Urbino, Italy; 3Department of Anatomy, Cell Biology and Physiology, Indiana University, Indianapolis, IN 46202, USA; fpin@iu.edu

**Keywords:** cancer cachexia, muscle wasting, mitochondria, regeneration, myogenesis, PGC-1α, metabolism

## Abstract

Cancer cachexia is a frequently neglected debilitating syndrome that, beyond representing a primary cause of death and cancer therapy failure, negatively impacts on patients’ quality of life. Given the complexity of its multisystemic pathogenesis, affecting several organs beyond the skeletal muscle, defining an effective therapeutic approach has failed so far. Revamped attention of the scientific community working on cancer cachexia has focused on mitochondrial alterations occurring in the skeletal muscle as potential triggers of the complex metabolic derangements, eventually leading to hypercatabolism and tissue wasting. Mitochondrial dysfunction may be simplistically viewed as a cause of energy failure, thus inducing protein catabolism as a compensatory mechanism; however, other peculiar cachexia features may depend on mitochondria. On the one side, chemotherapy also impacts on muscle mitochondrial function while, on the other side, muscle-impaired regeneration may result from insufficient energy production from damaged mitochondria. Boosting mitochondrial function could thus improve the energetic status and chemotherapy tolerance, and relieve the myogenic process in cancer cachexia. In the present work, a focused review of the available literature on mitochondrial dysfunction in cancer cachexia is presented along with preliminary data dissecting the potential role of stimulating mitochondrial biogenesis via PGC-1α overexpression in distinct aspects of cancer-induced muscle wasting.

## 1. Pathogenesis of Cancer Cachexia

Cancer is one of the most relevant causes of death all over the world and its incidence is estimated to markedly increase globally in the next two decades (http://who.int/gho/database/en (accessed on 17 August 2021). Patients affected by cancer very frequently develop a complex multifactorial syndrome defined as ‘cachexia’. This is an urgent unmet medical need. Indeed, about 90% of cancer patients are at risk of developing this syndrome, with a mortality rate that can reach up to 80% [1]. Cachexia markedly complicates the management of cancer patients, reducing quality of life, tolerance to treatments, and survival rates, the more so when anticancer treatments are adopted [2]. Consistently, health care costs are higher in cachectic than in non-cachectic patients [3].

Loss of body weight and skeletal muscle mass, associated or not with reduced adipose tissue, altered metabolism, and low-grade chronic inflammation are hallmarks of cachexia [4]. Muscle wasting mainly results from an altered regulation of protein synthesis and degradation rates, the latter generally prevailing over the former. Although protein hypercatabolism is one of the main features of cachexia, specifically interfering with the different intracellular proteolytic systems is only marginally effective in protecting against cancer-induced loss of muscle mass and function [5,6].

In the last years, a number of studies focused on the possibility that altered energy metabolism could be the trigger of muscle wasting. Indeed, metabolomic analyses have shown that energy and protein metabolism are markedly perturbed in the skeletal muscle and in the liver of tumor-bearing animals [7,8]. Consistently, hypermetabolism associated with altered energy balance has been detected in about 50% of newly diagnosed cancer patients [9]. Particularly relevant to the impaired energy metabolism is the occurrence of mitochondrial dysfunction, resulting from the combined alteration of organelle biogenesis, dynamics, and disposal. Indeed, a considerably high number of studies reported perturbations of the mitochondrial system in the skeletal muscle of tumor hosts [10]. As an example, both the oxidative capacity and the production of ATP are impaired in the muscle of animals bearing cachexia-inducing tumors [11], a situation that is further exacerbated by chemotherapy [12]. Such mitochondrial alterations likely contribute to muscle protein hypercatabolism [13]. Indeed, in experimental cancer cachexia, the former precede the latter [14]. Consistent with data obtained on experimental models, a study reported that about 50% of hypermetabolic cancer patients do not report any body weight loss and impairment of performance status [9].

On the whole, the observations above suggest that mitochondria can become a potentially useful therapeutic target to tackle cachexia. Along this line, exercise training is one of the most explored strategies. While being quite a simple option, the benefits of exercise on mitochondrial health can be achieved only when training is systematic, which can be unachievable in cancer patients due to exercise intolerance as a consequence of co-morbidities and/or fatigue. In the last decade, drugs mimicking exercise have been developed and investigated [15]. On the one side, this pharmacological approach overcomes patient exercise intolerance, yet on the other it does not completely replace the beneficial effects of training. In addition to exercise mimetics, several drugs able to improve energy metabolism by indirectly impinging on mitochondria have been investigated for their protective action in experimental cancer cachexia [10]. In this regard, quite recent data provide evidence that the exposure of tumor-bearing mice to the Szeto-Schiller peptide (SS-31), a drug able to directly target mitochondria, results in improved body weight loss, restoration of muscle succinate dehydrogenase (SDH) activity and ATP levels, and preservation of mitochondrial mass (see the next chapter for further details) [8].

Beyond directly targeting mitochondrial function, other exercise-mimicking strategies have been tested in preclinical cancer cachexia models, potentially impacting on mitochondrial health. As previously described, muscle wasting results from excessive protein degradation, although autophagy degradation may be halted by saturation or exhaustion of the system. Consistently, pharmacological stimulation of autophagy using rapamycin or AICAR (a known PGC-1α inducer) counteracts C26-induced muscle loss and atrogene induction [16]. Additionally, antioxidants have been extensively tested for their potential anti-cachexia activity, acting on the one hand by reducing the oxidative stress (thus improving mitochondrial function) and slowing down inflammation, on the other hand. Among the mostly studied antioxidants, resveratrol [17] and quercetin [18] have produced more positive results. The other way round, the use of antioxidants in cancer patients is not considered free from potential side effects, also in terms of tumor growth promotion [19], limiting their use in multimodal anti-cachexia clinical trials.

In addition to the altered energy metabolism, and likely interconnected with the latter, delayed muscle regeneration has been shown to take place in both tumor-bearing animals and cancer patients [20,21,22,23], with the transcription factor NF-κB [18] and enhanced activation of ERK stress kinase playing a role [17]. Similarly, overexpression in myogenic precursors of Twist1, a transcription factor associated with the malignant progression of several tumors, or of the zinc transporter ZIP14, is associated with impaired differentiation and muscle atrophy in preclinical models of cancer cachexia [24,25]. While it is widely accepted that mitochondria play a crucial role in maintaining muscle metabolic fitness, it is still unclear if muscle metabolic homeostasis and regenerative capacity can be reciprocally regulated. A profound metabolic reprogramming, including the increase of mitochondrial mass, occurs in myogenic precursor cells after their activation [26,27], while during regeneration, mitochondrial function is progressively restored to normal levels [28]. Taken together, these observations suggest that energy metabolism, mitochondrial function in particular, is relevant to correctly achieve muscle regeneration. Consistently, the differentiation potential of human myogenic precursors can be enhanced by overexpression of peroxisome proliferator-activated receptor gamma coactivator 1α (PGC-1α), a co-transcription factor working as a master regulator of mitochondrial biogenesis [29]. Along this line, our preliminary observations show that myogenic precursors isolated from transgenic mice with muscle-specific overexpression of PGC-1α (MCK-PGC-1α mice) and implanted into the muscle of wild-type (WT) animals fuse with existing myofibers and promote an oxidative phenotype shift more efficiently than cells obtained from non-transgenic animals (unpublished data).

## 2. Mitochondrial Alterations and Mitochondria-Targeted Anti-Cachexia Interventions

The unbalance between protein degradation and protein synthesis is one of the major culprits of skeletal muscle loss during cancer [30]. Different lines of evidence demonstrated that altered energy metabolism may potentially trigger or synergize with impaired protein turnover to affect muscle mass and function. Indeed, alterations of metabolites associated with glycolysis, TCA cycle, and fatty acid metabolism were detected in the skeletal muscle of tumor-bearing animals and even worsened in the presence of chemotherapy [31]. The metabolome unbalance found in the skeletal muscle was also corroborated by the alteration of the mitochondrial homeostasis in different preclinical models of cancer cachexia [31,32,33]. In this regard, protein levels of PGC-1α are found to be both decreased or unchanged in the skeletal muscle of animals carrying different types of cachectic tumors, and under different types of chemotherapeutic regimens [11,14,31,32,33,34,35]. Although a reduction of PGC-1α is not always detected in cachectic animals, mitochondrial alterations found in cancer cachexia cover a broad spectrum, from the organelle biogenesis and correct morphology to the disposal systems, ultimately affecting their quantity and function. In this regard, mitochondria are able to respond to muscle cues, remodeling their morphology through two different processes collectively defined as mitochondrial dynamics, namely mitochondrial fusion and fission [36]. In the muscle of tumor-bearing mice, the levels of mitochondrial dynamics indicators, such as Mfn2, Opa1, Fis1, and Drp1, are downregulated [11,31,32,37], suggesting an impairment of mitochondrial adaptation to muscle homeostatic changes. Together with altered mitochondrial biogenesis and dynamics, the disposal of mitochondria (mitophagy) is also affected in cachectic skeletal muscle, as suggested by the increase of bulk autophagy flux and of the mitophagy marker BNIP3 in the skeletal muscle of C26-bearing mice. In the same study, the lipidated form of LC3 II was found to be upregulated in mitochondrial-enriched fractions of cachectic muscles [13], suggesting an increase of molecular tagging of mitochondria towards degradation. In addition, another two independent reports demonstrated an increase in mitophagy-related proteins, such as PINK1 and Parkin, further suggesting an activation of mitochondrial degradation in the skeletal muscle of C26-bearing mice [11,34]. The alteration of mitochondrial biogenesis and disposal are in accordance with the common finding of reduced mitochondrial mass, commonly measured by quantifying protein or transcript levels of mitochondrial proteins, such as cytochrome c (cyt c), SDHA, COX IV, TFAM, and TOMM20, as well as the content of mitochondrial DNA [11,31,32,38]. The disrupted mitochondrial homeostasis found in cachectic animals results in reduced mitochondrial function combined with alterations in the levels of energy-related metabolites [8,11,12]. Indeed, different studies show an impairment of mitochondrial respiration ex vivo or in isolated mitochondria from cachectic muscles [8,13,14]. Altered mitochondrial function also leads to increased ROS production and oxidative stress, further promoting mitochondrial damage [10]. In this regard, the mitochondria-specific phospholipid cardiolipin is particularly affected by oxidative stress [39]. Cardiolipin is an essential constituent of the mitochondrial membrane and is crucial for proper mitochondrial function [40]. Our recent results demonstrated that cardiolipin and its precursor phosphatidylglycerol were reduced in C26-bearing mice, suggesting an intrinsic alteration of the mitochondrial membrane contributing to reduced mitochondrial function [8]. Another recent study confirmed these results by showing alteration of the muscle phospholipid profile, including cardiolipins, in the skeletal muscle of urothelial carcinoma-bearing mice [41]. Importantly, mitochondrial alterations have been validated in the skeletal muscle of patients affected by cancer cachexia. As compared to weight-stable cancer patients, cachectic patients showed increased muscle expression of Fis1 together with altered mitochondrial morphology [42]. Collectively, these studies implicate mitochondrial dysfunction as a key contributor to muscle wasting.

As briefly introduced above, exercise is a pleiotropic and non-pharmacologic approach that, among many different targets, modulates mitochondria homeostasis. Countless evidence exists on how both aerobic and resistance exercise, or the combination of both, are effective in counteracting muscle wasting in cachectic animals [11,16,33,34,43,44,45]. The effect on muscle mass and function by exercise may rely in part on the increase of mitochondrial quantity and quality. In this regard, increased PGC-1α levels were observed in exercised C26-bearing mice treated or not with chemotherapy [11,34,45]. In Apc+/Min and C26-bearing mice, exercise induces an increase of mitochondrial mass, suggested by increased levels of cyt c, SDHA, and COX IV [11,34,43,45]. Exercise impacts on mitochondrial homeostasis in chemotherapy-treated cachectic mice also by increasing the gene expression of the mitochondrial fusion protein Mfn2 and by suppressing mitophagy-related genes [11]. Exercise-dependent improvement of the whole mitochondrial machinery results in improved mitochondrial performance, indicated by increased SDH activity and ATP content in the skeletal muscle of cachectic mice [11,34]. The beneficial effects of exercise training on tumor-induced mitochondrial derangements provided an overview of different targets that may be modulated to improve mitochondrial homeostasis and possibly muscle wasting in cancer cachexia.

Despite the evidence supporting the relevance of cancer-induced disruption of skeletal muscle mitochondrial homeostasis, few studies have aimed to directly target mitochondrial alterations in cancer cachexia, so far. The above-mentioned mitochondrial-targeting peptide Szeto-Schiller-31 (SS-31) proved effective in reversing mitochondrial dysfunction in several diseases [46,47,48,49,50,51]. The efficacy of this compound relies on its ability to bind cardiolipin, protecting this key phospholipid from oxidation-induced damage, thus promoting mitochondrial function [39]. In C26-bearing mice treated with chemotherapy, SS-31 had protective effects against body and muscle wasting, along with improved mitochondrial activity [8], associated with decreased mitophagy. The metabolomic profile of SS-31-treated cachectic mice also revealed an increase of ATP levels, implying a partial restoration of energy metabolism in the skeletal muscle [8]. Interestingly, systemic administration of SS-31, beyond targeting the skeletal muscle, positively influenced both the liver and circulating metabolome by rescuing the levels of specific metabolites that are decreased in cachectic animals, including glucose and glycogen [8]. In addition, the depletion of liver glutathione observed in chemotherapy-treated C26-bearing mice was reversed by SS-31, suggesting an increase in the antioxidant capacity [8]. The improvement of the liver metabolome by SS-31 directs the attention to the relevance of mitochondria to other organs besides the skeletal muscle in correcting the systemic metabolic derangements induced in cancer cachexia. Other mitochondria-targeted compounds have been investigated for the treatment of cancer-induced cachexia. Among them, trimetazidine (TMZ) targets β-oxidation by inhibiting the 3-ketoacyl-CoA thiolase activity in the mitochondrial matrix [52]. TMZ shifts glucose metabolism towards mitochondrial respiration, reducing the lactate levels and increasing oxygen consumption [53]. In cachectic mice, TMZ prompted a metabolic shift towards oxidative muscle fibers, increasing SDH activity, and protein levels of PGC-1α and other mitochondrial-related proteins. Such improvements result in increased muscle mass and strength as compared to untreated tumor-bearing animals [38].

The complex metabolic impact of exercise or exercise mimetics renders the identification of the key genes controlling mitochondrial and metabolic dysfunction in cancer cachexia difficult. Additionally, only a few studies have implemented genetic approaches to target mitochondria or metabolism-related genes to counteract cachexia. A recent study investigated the role of the pyruvate dehydrogenase kinase 4 (PDK4), a mitochondrial enzyme critical for regulating cellular energy metabolism [54]. PDK4 inhibits the activity of the pyruvate dehydrogenase complex, responsible for the conversion of pyruvate into acetyl-coenzyme A, thus regulating the influx of the three-carbon molecule into the TCA cycle [54]. PDK4 has been found to be upregulated in the skeletal muscle of cachectic mice and this alteration was associated with reduced PDH and SDH activity [55]. In addition, the pharmacologic activation of PDK4 in healthy animals phenocopied the muscle wasting features as well as the alteration of mitochondrial-related proteins observed in cancer-induced cachexia [55]. Interestingly, in vitro deletion of PDK4 protected cultured myotubes from tumor-conditioned medium-induced atrophy, suggesting PDK4 as a target to improve the mitochondrial homeostasis and the energetic status of cachectic mice [55]. An overview of the currently investigated mitochondria targets is presented in Figure 1.

## 3. Role of Muscle PGC-1α in Cancer Cachexia

Starting from the observation that some of the exercise-based interventions and exercise mimetic effects in tumor-bearing animals are based on the ability to stimulate the expression and the activity of the master regulator of mitochondrial biogenesis PGC-1α, several studies have focused on the impact of this transcriptional coactivator on muscle phenotype, both in health and in wasting conditions, including cancer cachexia. Almost 20 years ago, Spiegelman’s group described for the first time PGC-1α preferential expression in oxidative fibers and characterized a transgenic mouse overexpressing PGC-1α specifically in the skeletal muscle (MCK promoter), showing a conversion towards slow-twitch muscle fibers [56]. On the contrary, myocyte-specific (Myogenin/MEF2 Cre) PGC-1α and β deletion impairs mitochondrial electron transport chain activity without impairing mitochondrial mass and physiological muscle function [57]. Despite being dispensable, PGC-1α mRNA rapidly decreases during atrophy induced experimentally by denervation, cancer cachexia in AH-130-bearing rats, streptozotocin-induced diabetes, and subtotal nephrectomy renal failure [58]. In the above-mentioned transgenic mice overexpressing PGC-1α, muscle mass loss and the induction of atrogenes induced by denervation and fasting were prevented via FoxO3 blockade [58], suggesting a mechanistic explanation for the anti-cachexia effect of exercise training. On the same line, PGC-1α overexpression was shown to prevent sarcopenia associated with aging or induced by hindlimb unloading [59,60].

Whether PGC-1α is also able to counteract cancer-induced muscle wasting is, however, still debated. Our previous observation showed that PGC-1α overexpression was either effective or not in preventing cancer-induced muscle wasting in young mice upon LLC or C26 tumor growth, respectively [11,33]. The protection observed in mice bearing the LLC tumor is in discordance with the data obtained by others [61] adopting apparently the same experimental conditions, where the lack of muscle wasting prevention may be potentially explained by the increased tumor growth observed in transgenic mice. Finally, an alternative splicing PGC-1α isoform (PGC-1α4) induced by resistance exercise was discovered [62] and muscle-specific overexpression determined muscle mass and strength increase, along with resistance to LLC-induced muscle wasting. The inconclusive observations may also depend on the fact that most preclinical proof of concept data come from studies using young animals, mainly for economic reasons, and not receiving chemotherapy, both conditions that limit the translational value of such models [63]. We considered such limitations and produced preliminary observations on the role of PGC-1α in an optimized preclinical model of cancer cachexia adopting both middle (14 months) and aged (22 months) mice bearing the LLC tumor and receiving doxorubicin (see Appendix A for details). Despite not protecting from body weight loss, PGC-1α overexpression counteracted muscle mass and strength loss in 14-month-old mice, while only improving muscle strength in 22-month-old mice. Despite the transgene being active only in the skeletal muscle, several parameters show a differential condition as compared to wild-type mice, including heart, white adipose tissue, and hematocrit, suggesting that boosting skeletal muscle mitochondrial biogenesis impacts on the whole body condition, deserving further investigation.

Although preliminary, our observations prompt new lines of investigation in the cancer cachexia field. The preservation of heart mass observed in transgenic mice is consistent with data obtained in experimental hearth failure [64,65], suggesting that PGC-1α overexpression could become a target to maintain exercise tolerance in cancer patients or at least normal daily physical activity, an essential point in the assessment of quality of life. Beyond heart function, PGC-1α could increase exercise tolerance by improving anemia. To date, no easy mechanistic explanation can be given to support this effect, although angiogenesis has been reported as one of the systemic effects of PGC-1α [66]. Increased hematocrit might be a consequence of enhanced muscle mitochondrial respiration and angiogenesis, in order to supply an adequate amount of oxygen to peripheral tissues. Overall, targeting PGC-1α in order to mimic exercise might be a promising strategy to contain the functional decline induced by both tumor growth and chemotherapy in cachectic patients.

## 4. Impact of Mitochondrial Dysfunction in Impaired Myogenesis during Cancer Cachexia

Muscle metabolism is mainly considered as the coordinated set of biochemical reactions occurring in adult muscle fibers, marginally considering the role of myogenic cell turnover and the impact of their metabolism on the overall muscle function and metabolism. Starting from the principle that the myogenic program is a highly energy-demanding process, it is possible that, beyond the tumor- and host-derived circulating factors that impair myogenesis [67], the alterations of energy metabolism present in cancer cachexia also impair muscle regeneration. The scenario is even more complex when considering that the muscle is not only composed of adult fibers and muscle-specific stem (satellite) cells, since other non-satellite muscle progenitor and accessory cells present in the muscle microenvironment have been causally associated with cachexia onset [68]. Both intrinsic (satellite cell) and extrinsic niche/muscle fiber dysfunctions may coordinately alter the myogenic program. The existence of an intrinsic satellite cell regeneration impairment has been recently questioned, since satellite cells derived from C26-bearing mice are increased as compared to healthy mice and retain their myogenic potential, growing and differentiating normally [22,69,70], suggesting that extrinsic factors, metabolism first, may determine the impaired myogenesis. The other way round, in all the above-mentioned papers, other accessory muscle cells were shown to be modulated in the muscle of C26-bearing mice, including innate immune cells, mesoangioblasts, and other interstitial cells, showing on the one side the canonical myogenic commitment (Pax7 expression, [22]) while, on the other, an adipocyte-like differentiation when cultured in vitro, suggestive of an aberrant metabolic and differentiative program [70]. The current hypothesis is that the alterations of the muscle microenvironment, mainly reflected by increased proinflammatory signals, do not allow satellite cell differentiation. However, inflammation might not be the only trigger, although very few studies have investigated how metabolic determinants in the stem cell niche impact on regeneration. The metabolic profile, the metabolism-associated gene signature, and the metabolic requirements of satellite cells in different myogenic states in young and old mice have recently been reported [71]. From this study it emerged that in aging, when the regeneration capacity is impaired, satellite cells have low oxidative phosphorylation and preferentially perform cytoplasmic glycolysis. Moreover, perturbing the ability to oxidize fatty acids in healthy mice, a process requiring efficient peroxisomes and mitochondria, impairs muscle regeneration [71]. The results are in accordance with previous ones showing the high metabolic demand of adult activated satellite cells, supported by higher mitochondrial activity and autophagy [26]. Mitochondrial function and energy availability in the satellite cell niche can thus represent limiting factors impairing the myogenic process during cachexia. Along this line, a recent report highlighted the importance of glutamine supplied locally from macrophages to support myogenesis after injury or rescuing the ischemia or age-related myogenic alterations [72].

Muscle disuse, due to both fatigue and bedrest, frequently occurring in cancer patients, was shown to stimulate satellite cell activation and a regenerative response [73]. Unfortunately, muscle disuse in the presence of cancer exacerbates muscle wasting, potentially via aberrant induction of myogenesis [74]. The satellite cell niche is strictly dependent on the type, amount, and organization of the collagen present in the extracellular matrix (ECM). During aging, the altered ECM drives a fibrogenic conversion of satellite cells at the expense of their ability to complete the myogenic program [75]. Such an effect is of particular relevance when considering the association between old age, cancer diagnosis, and cachexia. ECM remodeling occurring either during effective regeneration or cachexia is due to a complex interaction involving innate and adaptive immune cells, fibroblasts, fibroadipogenic precursors (FAPs), and vessel associated pericytes [76], making it clear that focusing only on proinflammatory mediators generates a reductionist view. The contribution of each cell type and signals released to the alterations of the complex orchestrated myogenic machinery is far from being fully elucidated, and will likely allow in the future the identification of new prospective therapeutic targets. The scenario is further complicated by recent evidence that the muscle of cachectic animals and cancer patients shows signs of impaired innervation [77,78,79]. Muscle denervation, either resulting from motor neuron or neuromuscular junction (NMJ) alteration, could thus represent the primary cause or at least contribute to both myofiber atrophy and regeneration in cancer cachexia. The other way round, sarcolemmal damage was reported in cancer cachexia [22] and recently associated with loss of the dystrophin glycoprotein complex interacting protein myocilin [80], providing another potential trigger for satellite cell activation.

In order to restore the myogenic process, distinct approaches have been proposed in the last years (Figure 2). MEK-ERK MAPK pathway inhibition, likely interfering with the proinflammatory cytokine signaling cascade, was the first approach that proved effective in preventing Pax7 accumulation and partially countering cachexia [21], also incidentally confirmed in cancer patients [81]. Potentially acting through a similar mechanism of downstream inflammatory mediators, NF-kB inhibition prevented Pax7-dependent inhibition of myoblast fusion [22]. Interfering with the cytokine milieu could thus impact on distinct aspects of cancer and cachexia, including impaired regeneration, as recently shown by interleukin (IL)-4 administration that counteracted cachexia and prevented the accumulation of satellite cells in C26-bearing mice, directly inducing myocyte differentiation [70]. Beyond cytokines, circulating extracellular vesicles and their microRNA cargo have also been implicated in impaired myogenesis [82] through toll-like 7 receptor (TLR7) signaling. Consistently, a TLR7/8/9 antagonist prevented Pax7 overexpression and cachexia onset [83]. Whether all these approaches rescue the myogenic program through a direct transcriptional regulation of the myogenic genes or other mechanisms is still unknown and the possibility that part of the action relies on cell metabolism is intriguing and will likely be clarified in future studies. On this line, the fact that physical exercise rescues muscle cell differentiation in tumor-bearing animals [84] suggests that metabolism modulation may provide an effective tool targeting cachexia from a new perspective. Another point in favor of such hypothesis is the presence in cachexia of myosteatosis and increased circulating free fatty acids that may produce lipotoxicity, leading to impaired muscle performance and regeneration [85]. For a long time, it has been demonstrated that satellite cell abundance is influenced by fiber metabolism, with the oxidative having more and the glycolytic less satellite cells [86]. The same authors showed that promoting a glycolytic to oxidative fiber shift leads to an increase in the satellite cell number. Such a mechanism also proved effective during cancer cachexia. Indeed, the above described TMZ anti-cachexia action, by promoting muscle oxidative metabolism via PGC1α overexpression, enhanced myogenesis in C26-bearing mice [87].

## 5. PGC-1α Overexpression Rescues Mitochondrial Content in Regenerating Muscles of Tumor-Bearing Mice

Although several lines of evidence suggest a connection between altered metabolism and impaired regeneration, it is still unclear whether transcription factors and coactivators involved in the control of muscle metabolism play a role in controlling muscle repair. As mentioned above, key mitochondrial players, such as PGC-1α, are found to be both unaffected or modulated among different preclinical cancer cachexia models and cachectic cancer patients. In details, chemotherapy-naïve animals bearing either the LLC or the C26 tumor present with no changes in PGC-1α levels [33], yet our observations presented here reveal that PGC-1α expression is significantly decreased in the skeletal muscle of C26-bearing mice only when a muscle injury has been inflicted (i.m. injection of BaCl_2_; Figure 3A,B, see Appendix A for methodological details). This new result suggests that PGC-1α repression might be partially accountable for defective muscle regeneration in cancer cachexia, with the muscle progenitor being the main cell type requiring PGC-1α expression to maintain skeletal muscle cellular turnover.

To further investigate this point, we tested whether forced overexpression of PGC-1α was sufficient to improve deficient regeneration in cancer cachexia by injuring the skeletal muscle of MCK-PGC-1α mice bearing the C26 tumor. The muscle architecture 14 days post-injury consisted of centrally nucleated myofibers (Figure 3C) with similar mean myofiber cross-sectional (CSA) area in both WT (64.1 ± 10.6% of uninjured muscle fibers) and MCK-PGC-1α (65.5 ± 27.5%) animals (Figure 3D), implying that PGC-1α overexpression does not accelerate muscle regeneration in healthy conditions. Consistent with previous reports [63], injured muscles of C26-bearing WT mice presented with delayed myofiber CSA recovery (32.3 ± 7.2%), which was not improved by PGC-1α overexpression (33.0 ± 13.6%; Figure 3D).

Although forced expression of PGC-1α was not sufficient to rescue myofiber maturation in cachectic animals, visual inspection revealed loss of pigmentation in injured muscles that was further aggravated in tumor bearers, suggestive of reduced mitochondrial content (Figure 3E). Macroscopic observations were supported by a decreased protein expression of cyt c in cachectic WT animals, indicating a decrease in mitochondrial mass (Figure 3F). On the contrary, injured muscles of cachectic MCK-PGC-1α mice presented with a marked brown coloring and spared levels of cyt c protein (Figure 3E,F), suggesting that PGC-1α overexpression preserves mitochondrial mass. On the whole, the present results indicate that PGC-1α overexpression is not sufficient to rescue muscle regeneration delay, although ameliorating the mitochondrial status, opening new possibilities to therapeutically modulate the metabolism in association with myogenesis-specific enhancers to impact on muscle regeneration in cancer cachexia.

## 6. Conclusions

Based on the vast amount of data from both preclinical models and cancer patients, muscle wasting in cancer cachexia is now considered as a complex metabolic syndrome, with specific hallmarks, only in part shared with other wasting disorders. The emerging role of mitochondrial and metabolic alterations along with the metabolism-targeted approaches, including the genetic PGC-1α overexpression described herein, will likely lead to the optimization of new therapeutic options for cancer cachexia. Similarly to the hallmarks of aging [88], we here propose a scheme (Figure 4) describing muscle atrophy during cancer cachexia as a multi-phasic process characterized by initiating causes (primary hallmarks focused on cancer cachexia etiology), followed by the adaptive responses activated in both muscle fibers and the stem cell compartment (antagonistic hallmarks), eventually leading to the cachectic phenotype (integrative hallmarks).

In the future, new tools allowing single-cell analyses, including mass spectrometry imaging and single-cell metabolomics, will clarify how the distinct muscle components, from adult muscle fibers to myogenic precursors and accessory cells, are affected by both tumor growth and chemotherapy, potentially allowing the identification of new prospective therapeutic targets for cachexia directly impacting on cell metabolism.

## Figures and Tables

**Figure 1 cells-10-03150-f001:**
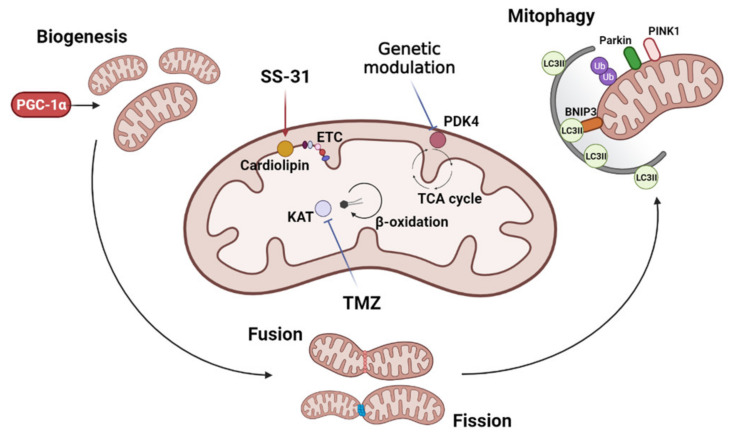
Mitochondria-targeted anti-cachexia interventions. Interventions focused on mitochondria to improve cancer and chemotherapy-induced muscle wasting may include the modulation of mitochondrial biogenesis, dynamics (fusion and fission), and mitophagy. Different strategies have been adopted to directly target mitochondria: SS-31 protects cardiolipin, enhancing mitochondrial activity and energy production through increased electron transport chain (ETC) efficiency; trimetazidine (TMZ) inhibits the 3-ketoacyl-CoA thiolase (KAT) activity, shifting the mitochondrial metabolism towards oxidative respiration and increasing mitochondrial function; the genetic deletion of pyruvate dehydrogenase kinase 4 (PDK4) enhances the influx of pyruvate into the TCA cycle, likely improving mitochondrial homeostasis. Image created with BioRender.com (accessed on 17 August 2021).

**Figure 2 cells-10-03150-f002:**
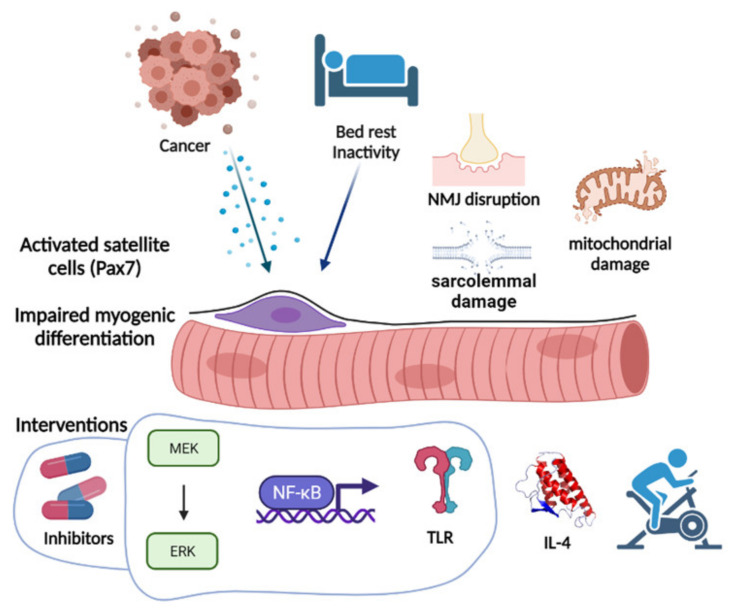
Cancer and the consequent inactivity impair the myogenic process, either directly via cytokine release or through several mechanisms including neuromuscular junction (NMJ) disruption, and mitochondrial and sarcolemmal damage. Several interventions have been proposed, mainly focused on signaling pathway inhibition, cytokine modulation, and exercise or exercise-mimicking metabolism modulators. Image created with BioRender.com (accessed on 17 August 2021).

**Figure 3 cells-10-03150-f003:**
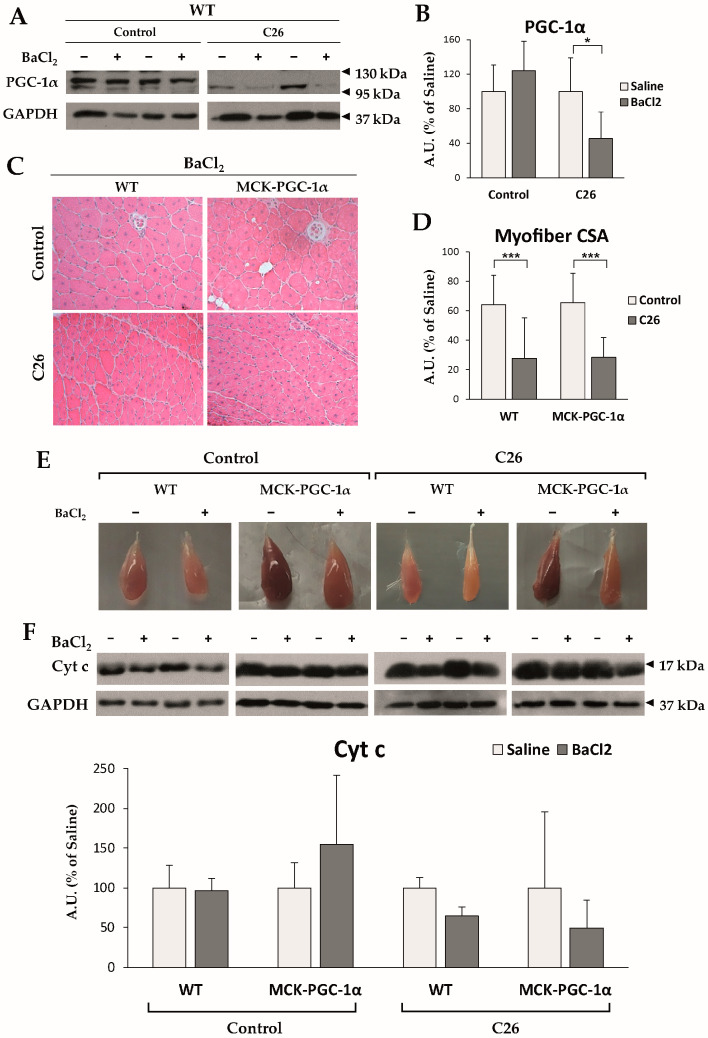
(**A**,**B**) Representative Western blotting bands and densitometric analysis of PGC-1α protein expression, normalized by GAPDH. Data are reported as % ± SD of the uninjured control group; (**C**,**D**) Representative H&E panels and CSA quantification of injured muscles of either healthy or cachectic WT and MCK-PGC-1α mice. Data are expressed as % ± SD of the uninjured WT control group; (**E**) Representative pictures of tibialis anterior muscles according to its specific experimental condition. (**F**) Representative Western blotting bands and densitometric analysis of cyt c protein, normalized by GAPDH expression. Data are reported as % ± SD of uninjured muscle in each experimental condition. For all panels, the significance of the differences was assessed by Student’s *t*-test for independent samples: * *p* < 0.05, *** *p* < 0.001.

**Figure 4 cells-10-03150-f004:**
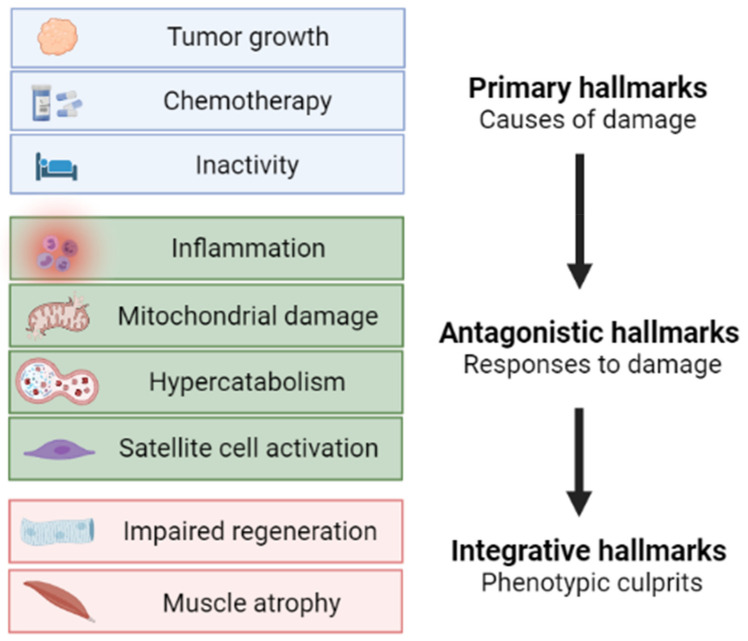
Proposed schematic pathogenesis of muscle wasting in cancer cachexia according to primary, antagonistic, and integrative hallmarks. Image created with BioRender.com (accessed on 17 August 2021).

## Data Availability

The data presented in this study are available on request from the corresponding author.

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
