# Peer review of "Mitochondrial Dysfunction in Cancer Cachexia: Impact on Muscle Health and Regeneration"

_cells, 2021, doi:10.3390/cells10113150_

Round 1

Reviewer 1 Report

The review article entiteld: "Mitochondrial dysfunction in cancer cachexia: impact on muscle health and regeneration" by Beltrà end colleagues is well written. The state of the art, as well as the data presented, provide a useful overview about the role of mitochondrial dysfunction in cancer cachexia.

However, the following suggestions can further improve the quality of the manuscript:

  • Lines 247-249: this part of the text describes data of muscle and heart mass of 14 and 22 months old mice, while figure 2B shows data from 12 and 24 months old mice
  • line 250: please specify here and in figure legend that "WT mice" means "WT bearing the LLC tumor mice"
  • Lines 266-274: according with the order of data reported in the text the graph of muscle strength should be moved before the one of hematocrit. Please, revise then also the figure legend
  • Line 393-394: can the differences observed in the pigmentation of muscles also due to their different content of myoglobin?

Author Response

Reviewer 1:

The review article entiteld: "Mitochondrial dysfunction in cancer cachexia: impact on muscle health and regeneration" by Beltrà end colleagues is well written. The state of the art, as well as the data presented, provide a useful overview about the role of mitochondrial dysfunction in cancer cachexia.

We are very grateful to the reviewer for the kind evaluation of our manuscript and we apologize for the mistakes found, that have been fixed in the revised version.

However, the following suggestions can further improve the quality of the manuscript:

Lines 247-249: this part of the text describes data of muscle and heart mass of 14 and 22 months old mice, while figure 2B shows data from 12 and 24 months old mice

The mistake has been fixed and the data have been moved to the appendix in order to lower the emphasis on original data as suggested by both Reviewer 2 and Academic Editor.

line 250: please specify here and in figure legend that "WT mice" means "WT bearing the LLC tumor mice"

The correction has been done, many thanks for the suggestion improving the clarity of the text.

Lines 266-274: according with the order of data reported in the text the graph of muscle strength should be moved before the one of hematocrit. Please, revise then also the figure legend

The correction has been done, many thanks for the suggestion.

Line 393-394: can the differences observed in the pigmentation of muscles also due to their different content of myoglobin?

In normal conditions, muscles with a predominant representation of oxidative myofibers are characterized by a darker pigmentation in comparison to muscles predominantly formed of glycolytic ones (e.g. soleus vs EDL). Oxidative muscles have increased mitochondrial density and higher expression of proteins involved in oxidative metabolism, including myoglobin. Consistently, the skeletal muscle of MCK-PGC-1α animals is characterized by a marked pigmentation, increased mitochondrial density and myoglobin expression (PMID: 12181572). Following this idea, the reduction in the pigmentation observed in injured muscles of C26-bearers suggested a delayed restoration of components central to oxidative metabolism, including a potential decrease in myoglobin levels (which could partially explain the observed phenotype). This initial hypothesis is supported by a reduced expression of Cyt c in injured muscles of WT cachectic mice, which is prevented by PGC-1α overexpression.

Reviewer 2 Report

In this manuscript, the authors described the current knowledge on mitochondrial dysfunction resulting in cancer cachexia together with their preliminary data obtained from authors on the potential role of stimulating mitochondrial biogenesis.

The authors present the mechanism of muscle wasting and cachexia caused by mitochondrial dysfunction and their therapeutic approaches. However, the mechanisms of mitochondrial malfunction, energy balance disruption, muscle loss, and the targets of therapeutic intervention are hard to follow in the context and need to be organized and reconstructed more. The insights based on significant recent findings regarding the mechanism, ideas, hypotheses are lacking compared to similar review articles dealing with mitochondria dysfunction, cancer, and muscle wasting. Moreover, many authors' previous papers are cited without explaining why these works are more important than others' previous findings and suggestions. Most of all, the authors' excessive amount of unverified preliminary results without mentioning the published noteworthy findings and scientific advances significantly undermine this manuscript's balance as a review paper.  

Author Response

Reviewer 2:

In this manuscript, the authors described the current knowledge on mitochondrial dysfunction resulting in cancer cachexia together with their preliminary data obtained from authors on the potential role of stimulating mitochondrial biogenesis.

The authors present the mechanism of muscle wasting and cachexia caused by mitochondrial dysfunction and their therapeutic approaches. However, the mechanisms of mitochondrial malfunction, energy balance disruption, muscle loss, and the targets of therapeutic intervention are hard to follow in the context and need to be organized and reconstructed more. The insights based on significant recent findings regarding the mechanism, ideas, hypotheses are lacking compared to similar review articles dealing with mitochondria dysfunction, cancer, and muscle wasting.

We thank the reviewer for the valuable assessment of our work. We definitely agree on the fact other reviews are better focused on the potential link existing between mitochondrial dysfunction and muscle wasting. In the present manuscript, the main attempt to differentiate it from the already published literature was made by exploring the importance of aspects still untouched by the other reviews, i.e. the role of PGC-1alpha modulation and the impact of mitochondrial dysfunction on impaired myogenesis in cancer cachexia. We tried to revise the manuscript taking into consideration this issue (see the red-tracked version), trying however not to change the focus from the main distinctive subjects.

Moreover, many authors' previous papers are cited without explaining why these works are more important than others' previous findings and suggestions.

As suggested by the Academic Editor, we tried to fix this point by reviewing again the literature and adding the references we considered consistent with our discourse. In the case we missed the point, we would be very grateful if the reviewer could suggest the milestones we omitted and we could easily integrate them in the text.

Most of all, the authors' excessive amount of unverified preliminary results without mentioning the published noteworthy findings and scientific advances significantly undermine this manuscript's balance as a review paper.

We partly agree with the reviewer on the fact that this manuscript is atypical, although not exceptional, in terms of presenting original data inside a main review manuscript. The other way round, the data presented here are aimed at supporting concepts that otherwise would be only speculative, given the absence of similar data in literature. Noteworthy, the idea of adding such data came after a discussion with the former Academic Editor, in the attempt to describe new potential research avenues and thus increasing the interest of the potential readers.  Regarding the unverified nature of the data, all the methodological information are available to the reader in the appendix A, thus any doubts on the scientific rigor can be discussed based on the described methods. Finally, in order to better balance the manuscript, the data originally presented in figure 2 moved to the appendix A and the description has been consistently reduced. In parallel, as previously mentioned, the parts dedicated to literature data in the main text have been expanded.

Reviewer 3 Report

I would like to much this type of article with a mixture between a typical review stile and a preliminary results, giving to the reader a complete overview of the role of mitochondrial dysfunction in the cachexia related with the cancer. Beltra M and coworkers in this manuscript with the title “Mitochondrial Dysfunction in Cancer Cachexia: Impact on Muscle Health and Regeneration” showed all the actual information about the state of the art of this aim and they integrated some of their preliminary data using an animal model where reflected the effects of skeletal muscle specific PGC-1α overexpression in LLC tumor-bearing mice with two different ages (14- and 22-months old) receiving doxorubicin. With the data showed open new lines of investigations in the field described in the article

The manuscript is well structured and is easy to read, however the point 3 (Line 288-359) with the title “Impaired myogenesis and targeted approaches in cachexia” describes a relationship between the myogenesis and the alterations of energy metabolism and other data but not the relation with the mitochondria. In my opinion this point in too long for the small relation that showed with the mitochondria dysfunction more than the energy metabolism but some information it is no relevant for this manuscript.

Minor comments:

Lines 268-271 the authors should be put in correct order the panels corresponding to figure 2, because the figure and the description in the text are not the same

Panel B of Figure 2 the age of animals is different than in the rest of panels and the description of the data in the text. Why is this?

The sex of animals used in this manuscript it is not described. There is same relation between the sex of the mice and the effect of cachexia and/or mitochondrial dysfunction?

In the part 4, Line 390-396 the author stablished a relation between CytC and mitochondrial mass, but I do not know this relation. The CytC is related with mitochondrial energy balance and with apoptosis, initiating the activation cascade of caspases but not with mitochondrial mass, could the authors explain me this connection?

In all the figures the authors used the Student’s t-test for analyzed the significance, but In my opinion should be more correct used a non-parametric test as Mann-Whitney

Author Response

Reviewer 3:

I would like to much this type of article with a mixture between a typical review stile and a preliminary results, giving to the reader a complete overview of the role of mitochondrial dysfunction in the cachexia related with the cancer. Beltra M and coworkers in this manuscript with the title “Mitochondrial Dysfunction in Cancer Cachexia: Impact on Muscle Health and Regeneration” showed all the actual information about the state of the art of this aim and they integrated some of their preliminary data using an animal model where reflected the effects of skeletal muscle specific PGC-1α overexpression in LLC tumor-bearing mice with two different ages (14- and 22-months old) receiving doxorubicin. With the data showed open new lines of investigations in the field described in the article.

We thank the reviewer for the appreciation of our work.

The manuscript is well structured and is easy to read, however the point 3 (Line 288-359) with the title “Impaired myogenesis and targeted approaches in cachexia” describes a relationship between the myogenesis and the alterations of energy metabolism and other data but not the relation with the mitochondria. In my opinion this point in too long for the small relation that showed with the mitochondria dysfunction more than the energy metabolism but some information it is no relevant for this manuscript.

This valuable reviewer comment is in line with the scanty data regarding the relationship between mitochondrial function and myogenesis. We have thoroughly revised this chapter and added literature focused on mitochondrial activity and myogenesis even out of the strictly cachexia field, including physiological regeneration and non-cancer dependent impaired regeneration. We hope that now the text is better delivering the message to the reader and better bound to the previous chapters.

Minor comments:

Lines 268-271 the authors should be put in correct order the panels corresponding to figure 2, because the figure and the description in the text are not the same

We apologize for the mistake, we have fixed this point as highlighted also in the response to reviewer 1.

Panel B of Figure 2 the age of animals is different than in the rest of panels and the description of the data in the text. Why is this?

Again, this was our fault and now the issue has been fixed.

The sex of animals used in this manuscript it is not described. There is same relation between the sex of the mice and the effect of cachexia and/or mitochondrial dysfunction?

The sex of the mice has been described in appendix A. Many thanks for noticing our negligence. The gender specific aspects would deserve a long and deep consideration, that unfortunately is out of the scope of the present review. We have never compared males and females in a same experiment with experimental cachexia-inducing cancers, however, we have observed impaired mitochondrial function in both sexes, thus supporting the idea that the mitochondria-target therapeutic strategies could be helpful in a sex-unbiased fashion, although further research is required to support such speculation.

In the part 4, Line 390-396 the author stablished a relation between CytC and mitochondrial mass, but I do not know this relation. The CytC is related with mitochondrial energy balance and with apoptosis, initiating the activation cascade of caspases but not with mitochondrial mass, could the authors explain me this connection?

As the reviewer has correctly stated, Cyt c is small protein associated with the inner membrane of mitochondrial that exerts an apoptotic function when released to the cytosol by promoting the apoptosome formation. However, when protein isolation is performed in whole tissue homogenate (combining cytosolic, nuclear and organelle proteins), “total amount” of Cyt c can be utilized as a marker of mitochondrial quantity/density as it follows a similar expression pattern as SDHA or COX-IV proteins (PMID: 30653354, 30823492).

In all the figures the authors used the Student’s t-test for analyzed the significance, but In my opinion should be more correct used a non-parametric test as Mann-Whitney

Data has been tested for normal distribution. Because all datasets represented in this manuscript have passed the normality test (Shapiro-Wilk test), the unpaired Student’s t-test (parametric) was used to determine if the means of two independent groups of data are significantly different.

Round 2

Reviewer 2 Report

The authors tried to respond to the reviewer's comments.
The revised manuscript is appropriate for publication in its present form.